# Percutaneous N-Butyl-Cyanoacrylate Embolization for Treating Ruptured Pancreaticoduodenal Aneurysm: A Case Report

**DOI:** 10.3390/medicina58101320

**Published:** 2022-09-21

**Authors:** Joo Yeon Jang, Jin Hyeok Kim, Tae Un Kim, Hwaseong Ryu, Tae Beom Lee, Je Ho Ryu, Ung Bae Jeon

**Affiliations:** 1Department of Radiology, Pusan National University Yangsan Hospital, School of Medicine, Pusan National University, Yangsan 50612, Korea; 2Division of Hepato-Biliary-Pancreatic Surgery and Transplantation, Department of Surgery, Pusan National University Yangsan Hospital, School of Medicine, Pusan National University, Yangsan 50612, Korea

**Keywords:** pancreaticoduodenal artery, aneurysm, percutaneous, NBCA, embolization

## Abstract

*Background and Objectives*: Pancreaticoduodenal artery aneurysms are rare visceral artery aneurysms. Interventional treatments, including transcatheter embolization, have an acceptable success rate. We report a case of ruptured pancreaticoduodenal aneurysm that was successfully treated with percutaneous N-Butyl-cyanoacrylate (NBCA) embolization after failed transcatheter embolization. *Materials and Methods*: A 53-year-old man presented to the emergency department with abdominal pain. Computed tomography (CT) revealed a ruptured aneurysm in the inferior pancreaticoduodenal artery (IPDA) with retrohemoperitoneum. The patient underwent percutaneous NBCA embolization after transcatheter embolization failure. *Results*: On CT, the pancreaticoduodenal aneurysm was completely embolized. No additional bleeding events occurred. *Conclusions*: Percutaneous NBCA embolization is safe and effective for treating patients with ruptured pancreaticoduodenal aneurysms after failed transcatheter embolization.

## 1. Introduction

Visceral artery aneurysms are rare, with a low incidence of 0.1% to 0.2% [1,2]. These aneurysms are uncommon compared to aortic or iliac aneurysms, but their rupture rate is substantially higher (25%) [3]. The splenic artery is the most frequent site of aneurysms and accounts for approximately 60–80% of visceral artery aneurysms [4]. The second most common site is the hepatic artery (approximately 20%), and only a certain percentage of visceral artery aneurysms occur at the superior mesenteric (5–7%), celiac (3–4%), pancreaticoduodenal (1–2%), or gastroduodenal artery (GDA) (1–2%) [4]. The pancreaticoduodenal artery (PDA) is a rare site for visceral artery aneurysms. However, the incidence of ruptured PDA aneurysms at presentation is twice that of GDA aneurysms, according to the literature. The aneurysmal rupture is correlated with a high mortality rate; therefore, the immediate treatment of PDA aneurysms is required. To the best of our knowledge, this is the first reported case of PDA aneurysm being successfully treated under emergent conditions by transabdominal ultrasound (US)-guided N-Butyl-cyano acrylate (NBCA) injection after failed transcatheter embolization.

## 2. Case Report

A 53-year-old man visited another hospital because of abdominal pain. He was referred to our emergency room after computed tomography (CT) was performed at our hospital. The patient was previously healthy and had no specific medical history. The initial blood pressure was 110/70 mmHg, with a pulse rate of 122 beats per minute. The laboratory values were as follows: hemoglobin 12.7 g/dL (13.5–17.5 g/dL); white blood cell count 14 k/μL (4–11 k/μL; platelet count 243 k/μL (140–400 k/μL; serum amylase 49 IU/L (22–80 IU/L); lipase 25 U/L (0–67 U/L); prothrombin time/international normalized ratio 1.02 (0.8–1.2); C-reactive protein (CRP) 0.09 mg/dL (0–0.5 mg/dL). The CT scan performed at the other hospital revealed retro-hemoperitoneum and an inferior pancreaticoduodenal artery aneurysm 7 mm in size (Figure 1A). In our hospital, a follow-up CT was performed, which showed that the retrohemoperitoneum had increased in extent, and the aneurysm had increased in size to 12 mm (Figure 1B) within a span of 5 h. Therefore, we thought this was a ruptured aneurysm, and emergent embolization was needed.

The patient was transferred to an angiography suite. After local anesthesia (lidocaine 2%), the right common femoral artery was punctured under ultrasound guidance using the Seldinger technique. Subsequently, a 5 Fr sheath (Terumo, Tokyo, Japan) was inserted. The celiac artery was selected using a 5Fr Rosch hepatic catheter (Cook, Bloomington, IN, USA). The aneurysm was not visible on the celiac artery angiography. However, it was seen on the superior mesenteric arteriography because of celiac stenosis (Figure 2A). For a better approach to the aneurysm, a 0.035-inch, 150 cm-long hydrophilic guidewire (Terumo, Tokyo, Japan) and a 5 Fr Cobra catheter (Merit Medical, South Jordan, UT, USA) were used. The aneurysm and the feeding artery were embolized with an NBCA (N-butyl cyanoacrylate; Histoacryl, B. Braun, Melsungen, Germany) and Lipiodol (Guerbet, Villepinte, Aulnay-sous-Bois, France) 1:3 mixture after superselection of the feeding branch of the inferior pancreaticoduodenal artery using a microcatheter (Asahi Tellus; ASAHI INTECC Co., Ltd., Seto, Japan) and micro-guidewire (Asahi Meister; ASAHI INTECC Co., Ltd.). Unfortunately, the aneurysm ruptured, and definite extravasation of the contrast media was observed on post-embolization angiography (Figure 2B). The patient complained of severe abdominal pain. In this situation, open surgery should be considered the first treatment option. The surgeon was on standby for the possibility of emergency surgery at the time. However, it is difficult to find the aneurysm in the surgical field of view. Furthermore, the surgery is associated with high morbidity and mortality. After discussion with the surgeon, we tried the percutaneous treatment of the aneurysm. Owing to the patient’s thin body habitus, the aneurysm was well visualized on transabdominal ultrasound (US). The procedure was performed under local anesthesia, and a 21G Chiba needle (PTC-21G1, A&A Medical Device, Seongnam, Korea) was inserted into the aneurysm sac through the anterior abdominal wall under US guidance (Figure 2C). Embolization was attempted again via the Chiba needle with a 1:3 mixture of NBCA–lipiodol. Final angiography revealed a complete exclusion of the aneurysm (Figure 2D).

After embolization, the patient’s vital signs were stable without any adverse events; therefore, he was discharged after one week. Two days later, a follow-up abdominal CT scan revealed a complete glue cast-filled aneurysm without any postprocedural complications. Approximately three days later, he visited our emergency room due to gastric outlet obstruction caused by an encapsulated hematoma with extrinsic compression of the third portion of the duodenum. Therefore, percutaneous catheter drainage of the hematoma was performed. After three weeks of supportive care, the patient was discharged. Follow-up CT performed approximately three months later showed an entirely embolized inferior pancreaticoduodenal artery aneurysm (Figure 3A) and a markedly shrunken retroperitoneal hematoma (Figure 3B).

## 3. Discussion

Visceral artery aneurysms are divided into true aneurysms and pseudoaneurysms. Unlike true aneurysms that contain all three arterial wall layers, pseudoaneurysms result from the disruption of the intimal and medial layers and do not have any arterial wall [5]. Typically, pseudoaneurysms show a more irregular margin and are surrounded by hematoma [4]. Pancreaticoduodenal artery (PDA) pseudoaneurysms are more frequent than true PDA aneurysms [6]. The most commonly reported causes of PDA pseudoaneurysms are inflammation (e.g., pancreatitis), surgery, and trauma [4]. However, our patient had no history of surgery or trauma. In addition, the serum amylase and lipase levels were normal. Therefore, it was considered a true aneurysm.

Interestingly, at least one-third of patients with PDA or gastroduodenal artery (GDA) aneurysms have celiac stenosis and occlusions. This suggests that increased collateral flow from the superior mesenteric artery (SMA) via the pancreaticoduodenal arcade is a risk factor for the development of these aneurysms in some cases [4]. However, the causative factor was not identified in the remaining patients. In our patient, celiac stenosis was noted on CT and confirmed on SMA angiography (Figure 2A). These findings support the hypothesis that this was a true aneurysm.

True PDA aneurysms are sporadic and account for only 1–2% of visceral artery aneurysms [4]. The most common symptom is abdominal pain (in both ruptured and unruptured aneurysms), as in our patient. The literature indicates that the incidence of ruptured PDA aneurysms at presentation is twice that of GDA aneurysms; however, there is no correlation between aneurysm size and rupture rate [6]. In the present case, the aneurysm ruptured at the time of detection.

Because of the significantly high mortality rate associated with aneurysmal rupture, the early treatment of PDA aneurysms is essential. Traditionally, surgical treatment is considered the first-line therapy option [7], but it is invasive and is associated with a high risk of infection, hemorrhage, and extended rehabilitation time. In addition, surgery is technically challenging due to the deep location of retroperitoneal PDA aneurysms, and up to 70% of PDA aneurysms are not found during surgery [6]. Nonsurgical management includes endovascular and percutaneous embolization techniques [8]. In emergent situations, endovascular treatment is currently the first choice, because of its lower invasiveness, faster bleeding control and lower mortality rate [9]. According to the vascular anatomy of the target lesion and the patient’s clinical condition, various embolic materials can be used for endovascular management, such as coils, gelfoam, thrombin, NBCA (N-butyl-cyanoacrylate), stent-grafts, and Amplatzer vascular plugs [10]. Endovascular transcatheter embolization was also attempted in this patient. However, it failed. Alternatively, a percutaneous approach was performed under US guidance. Finally, successful embolization was achieved with NBCA.

Therefore, we report a case of an extremely rare PDA aneurysm that was treated with percutaneous NBCA embolization.

## 4. Conclusions

PDA aneurysms are rare visceral artery aneurysms with a relatively high rupture rate. In the case of failed endovascular management of PDA aneurysms, percutaneous embolization can be an appropriate treatment option in some cases, instead of open surgery.

## Figures and Tables

**Figure 1 medicina-58-01320-f001:**
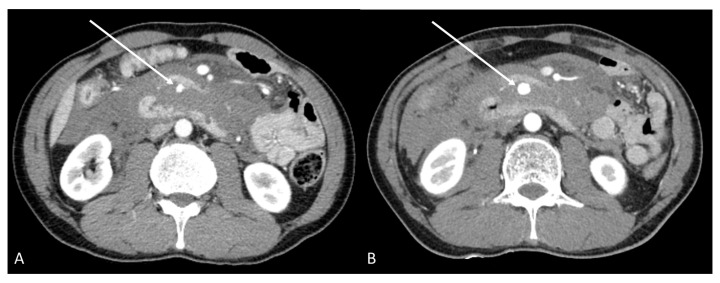
(**A**) On the computed tomography (CT) scan performed at the other hospital, a highly attenuated region of fluid collection was seen around the pancreas, small bowel mesentery, and the anterior pararenal space, and was considered retro-hemoperitoneum. In addition, the sac-like enhancing lesion seen near the uncinate process of pancreas, 7 mm in size, is the aneurysm (arrow). (**B**) On follow-up CT, the retro-hemoperitoneum increased in extent, and the aneurysm grew to 12 mm in size.

**Figure 2 medicina-58-01320-f002:**
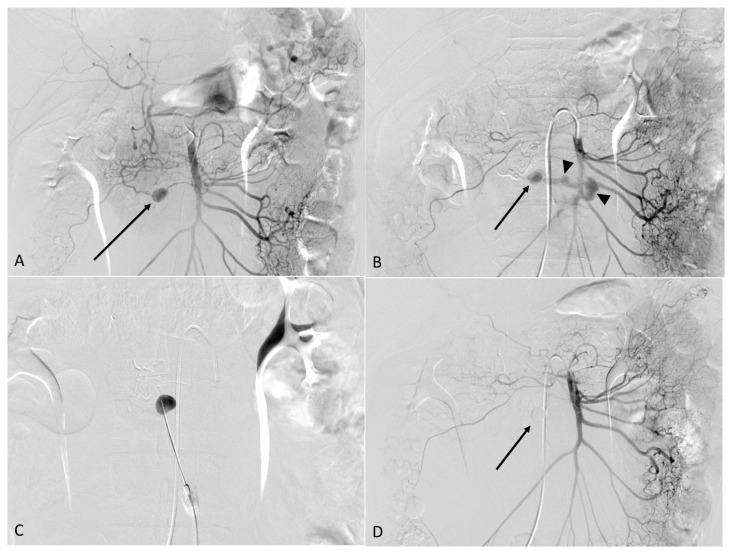
(**A**) The inferior pancreaticoduodenal artery aneurysm (arrow) is visible on the superior mesenteric arteriography. In addition, the hepatic artery was visualized through collateral blood supply via the pancreaticoduodenal arcade. This phenomenon can be seen in the case of celiac stenosis. (**B**) After the transcatheter embolization with the N-Butyl-cyanoacrylate (NBCA)–lipiodol mixture, the rupture of the aneurysm (arrow) with consequent contrast extravasation (arrowhead) was seen on angiography. (**C**) The aneurysm was punctured directly using a 21G Chiba needle under ultrasound guidance. (**D**) After the percutaneous NBCA embolization, the aneurysm disappeared on angiography.

**Figure 3 medicina-58-01320-f003:**
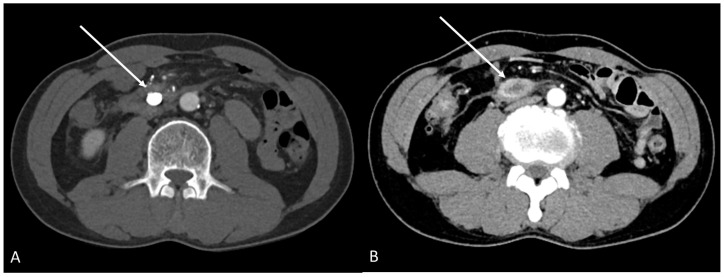
(**A**) The inferior pancreaticoduodenal artery aneurysm (arrow) was completely embolized on the latest CT image. (**B**) In addition, the retroperitoneal hematoma markedly decreased and was measured to be about 2 cm in size (arrow).

## Data Availability

Not applicable.

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
