# Peer review of "Percutaneous N-Butyl-Cyanoacrylate Embolization for Treating Ruptured Pancreaticoduodenal Aneurysm: A Case Report"

_medicina, 2022, doi:10.3390/medicina58101320_

Round 1
Reviewer 1 Report
A nice, concise report of a novel technique to deal with a potentially lethal situation.
I would simply ask the authors about their initial choice of embolic material to try to treat the aneurysm. Why not start with coils?
Author Response
Thank you for your comment. In an emergent situation like our case, NBCA can embolize the aneurysm in less time than using a coil. In other words, when using a coil, it takes some time for thrombus formation. On the other hand, if using NBCA, immediate embolization can be achieved. It also depends on the preference and familiarity of the operator. For these reasons, we chose NBCA for the initial embolic material.
Reviewer 2 Report
The authors report a case report on a ruptured visceral aneurysm.
The percoutaneus embolization is performed in some cases by others authors in non ruptured aneurysms.In the introduction the authors should better expose the problem of the ruptured visceral aneurysms.
In the description of the case it is singular the increment of the sac in few hours.The author should explain this. In the description the authors assert that there was an high risk of rupture but by TC the rupture was evident immediately.
The percoutaneus embolization in this case was very dangerous. I think that an open approach was more appropriate. The Authors should explain better this choice.A great dose of fortune was requested to obtain a good results. The Authors sould discuss these problems.The conclusions are not appropriate.A single case cannot lead any conclusion.The authors should assert that only in extreme cases such an approach can be attempted.In the maiority of these cases, after a failure of catheter embolization, an open operation should be preferred. In the reference the following paper could be quoted :single center experience in the treatment of visceral artery aneurysms by Martinelli O. et all, Annals of vascular surgery 2019.
Round 2
Reviewer 2 Report
the text has been improved and although the scientific interest is moderate it can be published in my opinion.Remain a doubt on the increment of the sac in few hours